# m^6^A RNA Modification: Technologies Behind Future Anti-Cancer Therapy

**DOI:** 10.3390/molecules30204091

**Published:** 2025-10-15

**Authors:** Kristina Shpiliukova, Artyom Kachanov, Sergey Brezgin, Vladimir Chulanov, Alexander Ivanov, Dmitry Kostyushev, Anastasiya Kostyusheva

**Affiliations:** 1Martsinovsky Institute of Medical Parasitology, Tropical and Vector-Borne Diseases, Sechenov University, Moscow 119991, Russia; kristi14035@gmail.com (K.S.); kachanov.av99@gmail.com (A.K.); seegez@mail.ru (S.B.); vladimir@chulanov.ru (V.C.); kostyusheva_ap@mail.ru (A.K.); 2Center for Precision Genetic Technologies for Medicine, Engelhardt Institute of Molecular Biology, Russian Academy of Sciences, Moscow 119991, Russia; aivanov@yandex.ru; 3Laboratory of Experimental Therapy of Infectious Diseases, Martsinovsky Institute of Medical Parasitology, Tropical and Vector-Borne Diseases, Sechenov University, Moscow 119435, Russia; 4Department of Infectious Diseases, Sechenov University, Moscow 119435, Russia; 5Faculty of Bioengineering and Bioinformatics, Lomonosov Moscow State University, Moscow 119192, Russia

**Keywords:** metastasis, outcomes, genetic technologies, NGS, targeted therapy

## Abstract

N6-methyladenosine (m^6^A) modifications are among the most prevalent epigenetic marks in eukaryotic RNAs, regulating both coding and non-coding RNAs and playing a pivotal role in RNA metabolism. Given their widespread influence, m^6^A modifications are deeply implicated in the pathogenesis of various cancers, including highly aggressive malignancies such as lung cancer, melanoma, and liver cancer. Dysregulation of m^6^A dynamics—marked by an imbalance in methylation and demethylation—can drive tumor progression, enhance metastatic potential, increase aggressiveness, and promote drug resistance, while also exerting context-dependent tumor-suppressive effects. Given this dual role, precise modulation of m^6^A levels and the activity of its regulatory enzymes (writers, erasers, and readers) represent a promising therapeutic avenue. In this review, we highlight recent advances in targeting m^6^A machinery, including small-molecule inhibitors, antisense oligonucleotides, and CRISPR/Cas-based editing tools, capable of both writing and erasing m^6^A marks and altering m^6^A methylation sites per se. By evaluating these strategies, we aim to identify the most effective approaches for restoring physiological m^6^A homeostasis or for strategically manipulating the m^6^A machinery for therapeutic benefit.

## 1. Introduction

N6-methyladenosine (m^6^A) is one of the most abundant and dynamically regulated post-transcriptional modifications in eukaryotic RNAs, and it is present on approximately 0.15–0.6% of all adenosines [1,2]. This epitranscriptomic mark plays crucial regulatory roles across various RNA types, including messenger RNAs (mRNAs), ribosomal RNAs (rRNAs), transfer RNAs (tRNAs), and non-coding RNAs, such as microRNAs (miRNAs), small nuclear RNAs (snRNAs), circular RNAs (circRNAs), and long non-coding RNAs (lncRNAs) [3]. By influencing RNA stability, translation efficiency, splicing patterns, and nuclear export [1,4], m^6^A serves as an underlying determinant of RNA fate and function in cellular processes. Furthermore, the m^6^A methylation profile not only influences the translation efficiency of mRNAs but also plays a crucial role in microRNA maturation and the function of circRNAs, which can regulate the expression of genes involved in innate immunity and other post-transcriptional pathways. Given this fundamental role in cellular physiology, m^6^A dysregulation is implicated in numerous diseases, including various types of cancer [5,6], neurological disorders [7], cardiovascular impairments [8], and inflammatory responses [9,10], making it a promising target for therapeutic interventions.

The plasticity of m^6^A methylation—governed by its ability to be added, removed, and recognized—enables dynamic gene regulation in mammalian cells, managed by conserved enzymatic machinery. Central to this system are the N^6^-adenosine methyltransferases (“writers”), exemplified by the METTL3-METTL14-WTAP multiprotein complex. Within this complex, METTL3 provides catalytic activity by interacting with the cofactor S-adenosylmethionine (SAM), while METTL14 enhances substrate specificity. Wilms tumor 1-associated protein (WTAP), and other cofactors, including VIRMA, ZC3H13, and RBM15/15B, facilitate complex stabilization and proper subcellular localization, collectively enabling site-specific methylation [11,12]. The resulting modification is subsequently decoded by recognition proteins (“readers”), such as YTHDC1-2 and YTHDF1-3, which contain conserved YTH domains, and insulin-like growth factor-2-binding proteins (IGF2BPs), that selectively bind methylated adenosines to following RNA processing [13]. For instance, YTHDC1 plays a key role in regulating splicing by interacting with SRSF3 and SRSF10 factors and in mRNA nuclear export via SRSF3 and NXF1 proteins. YTHDC2 stimulates translation efficiency or promotes mRNA degradation by recognizing m^6^A-enriched regions of specific transcripts. Meanwhile, the cytoplasmic YTHDF paralogs form a functional network: YTHDF1 enhances translation, YTHDF2 promotes mRNA decay by recruiting the CCR4–NOT deadenylase complex, and YTHDF3 cooperates with either YTHDF1 or YTHDF2 to facilitate translation or decay, respectively [14,15]. Similarly, IGF2BP reader proteins stabilize target transcripts by binding m^6^A-modified sites and protecting them from degradation [16]. In addition to writers, demethylases (“erasers”), such as FTO and ALKBH5 (alkB homolog 5), dynamically remove m^6^A marks via Fe(II)- and α-ketoglutaric acid-dependent oxidative demethylation, establishing reversible epitranscriptomic control [17]. Together, the triple system provides epigenetic flexibility to the transcriptome, with m^6^A deposition/removal and reader-mediated decoding targeted at maintaining m^6^A modification balance and regulating RNA metabolism.

Moreover, m^6^A modification is not uniformly distributed across RNA transcripts [18]. Genome-wide mapping reveals that the writer complex most frequently installs m^6^A modification within the consensus RNA motif RR-AC-H (R—purine—A/G, H—A/C/U) with enrichment near terminal exons with stop codons and in 3′-UTRs—positioning that strategically influences mRNA decay, gene expression, and cell survival and death [4,11,18]. Determining the precise positions of m^6^A methylation is a fundamental challenge in epitranscriptomics, as the functional outcome depends critically on the modification’s location within a transcript. One of the first methods for profiling RNA methylation was MeRIP (methylated RNA immunoprecipitation). This technique uses m^6^A-specific antibodies to capture methylated RNA fragments, enabling transcriptome-wide assessment of m^6^A distribution [19,20]. Following the development of MeRIP, several other immunoprecipitation-based technologies were created, including miCLIP and PA-m^6^A-seq [19]. However, these methods still typically fail to achieve single-nucleotide resolution and require substantial amounts of input RNA. Subsequently, the m^6^A-SAC-seq and picoMeRIP-seq systems were developed, specifically enabling m^6^A mapping at single-base resolution (m6A-SAC-seq) and profiling in single cells (picoMeRIP-seq) while requiring substantially less input RNA for analysis [21]. To visualize the methylation profile of individual transcripts in situ, one promising method is the TARS assay [22]. This approach enables the determination of both qualitative and quantitative parameters of m^6^A at specific adenosine sites within RNA in single cells. The utility of TARS was demonstrated by mapping distinct methylation sites on the lncRNA *MALAT1* in HeLa cells [22]. The determination of site-specific methylation profiles in individual transcripts can elucidate their physiological functions at the cellular level.

Disrupting the normal function of the m^6^A-related enzyme complex, which is vital for cell function, may lead to the formation of solid tumors as well as confer drug resistance in cancer, pointing to the fundamental role of m^6^A methylation in carcinogenesis [23]. This epigenetic mark exhibits a context-dependent duality: while high levels of METTL3-mediated methylation promote cell proliferation and metastasis and inhibit apoptosis in breast cancer (BC) [24], they are also implicated in non-small cell (NSCLC) and small-cell (SCLC) lung cancers. In contrast, METTL14 overexpression is associated with suppressed metastasis in hepatocellular carcinoma (HCC) [25]. Due to m^6^A’s involvement in multiple aspects of cancer biology—such as cell cycle regulation, epithelial–mesenchymal transition (EMT), angiogenesis, metastasis, immune response, and therapy resistance [26]—a detailed investigation of the functions of this epitranscriptomic modification in specific cancer types is critical for developing effective and personalized treatment strategies. In current clinical practice, the dynamic profile of m^6^A RNA modification, along with the expression levels of its regulatory proteins, shows promise as a prognostic biomarker and is directly linked to outcomes in various cancer types, e.g., elevated expression of YTHDF1 and IGF2BP2 correlates with worse overall survival (OS) in HCC [27]. However, the clinical translation of m^6^A regulators as therapeutic targets faces several challenges, including the complexity of developing targeted therapies due to the widespread nature of RNA modifications, their dual roles in essential cellular processes, potential toxic effects, and delivery difficulties. Despite these challenges, several therapeutic approaches are under investigation. Small-molecule inhibitors and antisense oligonucleotides are currently in early stages of preclinical and clinical development. In contrast, CRISPR/Cas-based technology, while promising, remains at the fundamental research stage due to challenges in reproducibility. In the following sections, we will provide a detailed analysis of the role of m^6^A and its regulators in the pathogenesis of key cancers alongside a discussion of potential therapeutic approaches.

## 2. m^6^A Methylation in Lung Cancer

Lung cancer is the most frequently diagnosed cancer worldwide, accounting for 11.6% of all cases in 2018, and is the leading cause of cancer-related mortality, responsible for 18.4% of total cancer deaths [28]. Its primary risk factor is tobacco smoking. It comprises two main histological types: non-small cell lung cancer, which accounts for over 85% of all cases, and the more aggressive small-cell lung cancer, which accounts for about >15% of all LC cases. Both subtypes are highly influenced by an imbalance of m^6^A methylation [29].

In NSCLC, dysregulation of m^6^A writers drives oncogenesis, with elevated METTL3 expression strongly correlating with poor prognosis. METTL3 overexpression enhances tumor cell viability, migration, and invasion through Bcl-2 pathway modulation and facilitates metastasis through *c-MYC*-mediated stabilization of m^6^A-enriched *LINC01006* and increased translation of oncogenes, including *EGFR* and *BRD4*—demonstrated in both in vitro and in vivo models [30,31]. METTL14, a member of the methyltransferase complex, promotes cell cycle progression, high metastatic potential, and resistance to cisplatin (DDP) chemotherapy via miR-19a-5p targeting (Figure 1) [29]. In contrast, elevated ALKBH5 demethylase activity reduces global m^6^A levels and improves stabilization of oncogenes (*MYC*, *SOX2*, and *SMAD7*) with the ALKBH5–m^6^A–YTHDF2 axis, driving an aggressive phenotype in KRAS-mutant NSCLC (Figure 1) [32]. Additionally, decreased expression levels of FTO combined with a high degree of reader proteins YTHDF1 и IGF2BP3 in clinical NSCLC tissues promote tumor progression, metastasis, and drug resistance by regulating the ESR1 transcript, which activates proliferative signaling cascades and provides resistance to endocrine therapies through ERα pathway initiation [33].

SCLC is characterized, as previously mentioned, by a more aggressive behavior than NSCLC due to rapid invasion and wider therapy resistance potential. As Sun, Y. et al. reported, METTL3 overexpression is directly related to chemoresistance to doxorubicin and cisplatin in both in vitro and in vivo SCLC models [34]. METTL3 was shown to drive DCP2 mRNA/protein degradation, influencing the resistance process through mitophagy and mitochondrial damage levels (Figure 1) [34,35]. Additionally, IGF2BP2 is overexpressed in cisplatin-resistant SCLC, where its effects on *Spon2* transcripts activate the IGF2BP2-SPON2 and PI3K/Akt proliferation signaling pathways (Figure 1) [36,37]. Accordingly, this pathway is likely to promote M2 macrophage polarization within the tumor microenvironment—pointing to IGF2BP2 inhibition as a promising strategy to restore DPP sensitivity in SCLC.

The dynamic landscape of m^6^A modification offers promising therapeutic prospects for lung cancer, particularly given its role in promoting oncogene expression, metastasis, and resistance to chemo- and immuno-therapy [38]. Findings indicate that m^6^A’s functional mechanisms are not restricted to specific cancer types but also vary considerably across lung cancer subtypes, necessitating extensive research and personalized therapeutic approaches. Consequently, further investigation is required to develop clinical epitranscriptomic prognostic models based on subgroup-specific m^6^A profiles and to establish novel anticancer drug targets regulating RNA fate.

## 3. Breast Cancer and m^6^A Metabolism

Breast cancer was the second leading cause of cancer-related mortality in women (6.6%) and ranks among the most prevalent malignancies worldwide, accounting for 11.6% of all cancer diagnoses in 2018 [28]. Early detection provides higher 5-year survival rates (~80%) in the luminal A breast cancer subtype. However, survival significantly declines in less common (particularly luminal B and HER2-enriched subtypes) and more aggressive molecular subtypes, with the poorest prognosis observed in triple-negative breast cancer (TNBC) [28]. One of the common reasons for hyperproliferation and the possibility of metastasis is a long-term oncogenic stimulus, which includes changes in methylation patterns between normal and cancer-related cells and the microenvironment.

Recent studies implicate numerous involvements of METTL3 in BC progression, comprising metastasis and tumor immune surveillance. Wan, W. et al. demonstrated a direct connection between METTL3 activity and PD-L1 mRNA and protein level regulation, which is a critical immune checkpoint protein that facilitates tumor immune escape, especially with TNBC cells (Figure 2) [39]. The inhibition of METTL3 leads to alterations in PD-L1 stability and production, reduced methylation grade, and prevents progression of xenograft tumors and BC-specific cell cultures. The effect of immunoregulation was additionally enhanced through a reader IGF2BP3 knockdown that was related to recognition and stabilization of m^6^A-enriched PD-L1 transcripts, promoting their methyltransferase-mediated degradation [39]. Moreover, METTL3 upregulation in BC tissues and cell lines impacts uncontrolled proliferation and invasion, and suppresses apoptosis by stabilizing Bcl-2 mRNA and affecting the *MALAT1*/miR-26b/HMGA2 axis, contributing to the development of adriamycin (ADR) chemoresistance via miR-221-3p/HIPK2/Che-1 pathway (Figure 2) [24,40,41]. However, there is conflicting evidence where low METTL3 expression as well as low methylation grade correlates with poor prognosis and metastasis modulation in TNBC by regulating of *COL3A1* gene expression [42,43].

Among the key oncogenic drivers in BC pathogenesis, m^6^A regulatory proteins, such as demethylases (ALKBH5) and reader proteins (e.g., YTHDF1, IGF2BP3), play vital roles through dynamic epitranscriptomic reprogramming. Under hypoxia, which is a common phenomenon of solid tumors, ALKBH5-mediated demethylation stabilizes NANOG transcripts, reactivating their function in the self-renewal trajectory and maintenance of the breast cancer stem cell (BCSC) pool (Figure 2) [44]. Concurrently, inhibition of YTHDF1 and YTHDF3 in BC cells decreased tumor progression and metastatic capacity by metabolic impairment, reducing PKM2 activity and EGFR modulation accordingly [45,46]. On the other hand, emerging evidence suggests METTL14 and ZC3H13 function as tumor suppressors in triple-negative breast cancer [47]. Its deficiency correlates with poor prognosis, driven by METTL14-mediated upregulation of oncogenic effectors LSD1 (a histone demethylase) and YAP1 (a Hippo pathway transcriptional coactivator) (Figure 2) [48]. While some of these regulators represent potential therapeutic targets for tumor suppression, further studies are required to confirm their unique participation and clinical advantage. Identifying targets that regulate the epitranscriptome and defining their specific functions holds significant potential for advancing early screening, diagnosis, and prediction of breast cancer.

## 4. Disruption of m^6^A Machinery in Melanoma

Melanoma represents a smaller proportion of all diagnosed skin cancer, but is responsible for about 75% of deaths related to skin cancer pathology [49]. Its incidence rate is increasing every year [50]. The key pathogenic drivers include ultraviolet (UV) radiation exposure, which is a primary external environmental carcinogen, along with internal risk factors such as genetic susceptibility (e.g., mutations in CDKN2A), phenotypic determinants (high nevus count) [50], and dysregulated epigenetic mechanisms, particularly involving RNA modifications like m^6^A methylation that influence oncogenic transcript stability.

The m^6^A methyltransferase METTL3 has emerged as a systematically implicated oncogenic driver in melanoma, while its overexpression controls tumor progression by increasing proliferation, migration, and invasion. Accordingly, METTL3 overexpression leads to an increase in the m^6^A activity that regulates specific genes to maintain cancer cells. Downregulated molecules include MMP2 and N-cadherin, both of which are necessary for melanoma metastasis and invasion (Figure 3) [51], TXNDC5, which increases cell proliferation potential [52], and UCK2, which enhances melanoma cell migration via the Wnt/β-catenin signaling pathway [53]. A similar role is performed by another m^6^A machinery protein type—FTO and ALKBH5. Elevated levels of these proteins promote a decline in m^6^A levels and stabilization of PD-L1, CXCR4, SOX10, and VEGF, resulting in tumorigenesis as well as insensitivity to immune checkpoint blockade (ICB) therapy in skin cancer, respectively (Figure 3) [5,54]. Moreover, knockdown of the reader YTHDF3 depletes proliferative, invasive, and migratory capacity by targeting LOXL3; the downstream targets of LOXL3 are MITF, TWIST1, SNAIL1, and PRRX1, which are involved in the EMT process [55].

Consequently, targeting these pathways, such as inhibiting the writer METTL3, erasers FTO/ALKBH5, or knocking down YTHDF3, represents a prospective therapeutic approach for melanoma progression and overcoming chemoresistance. Additionally, restoring m^6^A homeostasis by changing the m^6^A enzymes’ activity may suppress tumor-related AKT, Hippo-YAP, and MAPK signaling axes, thereby inhibiting melanoma growth [56].

## 5. Disrupted m^6^A Networks in Hepatocellular Carcinoma

Hepatocellular carcinoma (HCC) is the primary liver cancer, being responsible for 90% of all other types of liver cancer, and is characterized by a high mortality rate (~95%) and an extremely low 5-year survival rate (<6.9%) [57]. It typically develops in the context of chronic liver disease, driven by major risk factors including hepatitis B (and hepatitis C) viruses (HBV/HCV), alcohol addiction, non-alcoholic liver impairments, and aflatoxin B1 impact [57]. Due mostly to late-stage diagnosis and well-developed resistance to standard chemotherapeutic regimens, new molecular pathogenesis patterns are critically needed for the treatment of HCC.

Among the numerous molecular variants implicated in HCC, m^6^A RNA modification factors may be one of the central aspects in the regulatory mechanisms of HCC. Several studies identified METTL3, the catalytic core of the m^6^A methyltransferase complex, as critically associated with the driving force of oncogenicity signaling pathways, such as JAK/STAT, PI3K/AKT, and Hippo, causing the worst prognosis [58,59,60]. The opposite effect of another core protein of methyltransferase complex METTL14 was observed in the context of metastatic ability and tumor recurrence in HCC models in vivo and in vitro [25]. Ma, J. et al. determined the suppressive function of METTL14 through interactions with DGCR8 and pri-miR126, which was also defined as a metastasis suppressor [25].

Demethylase ALKBH5 also demonstrates HCC inhibition capacity decreasing LYPD1 mRNA stability and reducing HCC proliferation and metastasis [61]. Additionally, eraser FTO has shown its dual role in several studies. On the one hand, the overexpression of FTO has been identified as an oncogene that triggers the demethylation of PKM2, changing the metabolism of cancer cells towards aerobic glycolysis (Figure 4) [62]. On the other hand, it was found that the reduced expression of FTO is associated with poor prognosis and shorter overall survival time [63]. However, the fundamental mechanisms underlying its role in HCC are not yet fully determined.

Reader m^6^A proteins, including YTHDF1-3 and IGF2BP1-3, are further strongly involved in HCC pathogenesis by regulating the stability, mRNA translation, and modulating tumor immune microenvironment [64]. IGF2BPs proteins (IGF2BP1-3) play a critical oncogenic role in HCC by specifically recognizing and enhancing the expression of key transcripts. IGF2BP3 directly binds to and stabilizes the long non-coding RNA LINC01138, promoting HCC proliferation and invasion (Figure 4) [65]. Furthermore, IGF2BPs stimulate the translation of numerous well-established oncogenic targets, including IGF2, HMGA2, MCM10, and MMP9 [65,66]. YTHDF1 and YTHDF3 mediate recovery of HCC stemness and drug resistance through NOTCH1 activation [67] and HCC progression, migration, and EMT by targeting EGFR/STAT3 and WNT/β-catenin signaling axis (Figure 4) [68]. However, YTHDF2 may contribute to the cancer stem cell phenotype through the YTHDF2/OCT4 pathway, and at the same time, it could act as an HCC suppressor, influencing the ERK/MAPK pathway and initiating EGFR mRNA decay [69].

Finally, m^6^A RNA modification impacts critical and context-dependent effects across various malignancies, including HCC, melanoma, lung, and BC. Imbalance in its machinery between key regulators, writers, erasers, and readers, could promote tumorigeneses by modulating RNA stability, gene expression, and RNA localization of molecules strongly involved in processes of proliferation, migration, invasion metastasis, and immune and drug resistance [70]. However, further research is essential to define the roles of these regulators within specific cancer types and subtypes, enabling the identification of context-dependent therapeutic targets. Current efforts focus on pharmacologically inhibiting or modulating specific components of the m^6^A machinery to disrupt these pro-tumorigenic RNA regulatory networks.

## 6. Current Strategies Targeting m^6^A

Targeting the m^6^A RNA modification machinery represents an emerging and rational therapeutic progression in oncology. The goal of molecular searching among m^6^A regulators is to detect their specific changes to disrupt pathogenic RNA metabolism and overcome limitations of current therapies, like chemoresistance.

### 6.1. Small-Molecule Targeting m^6^A Regulation Factors

As mentioned earlier, dysregulation of m^6^A machinery frequently involves the elevated activity of specific modifiers (writers, erasers, and readers) in processes driving carcinogenesis. Consequently, the development of small-molecule inhibitors designed to normalize this aberrant activity (particularly by suppressing hyperactivated erasers or readers) has emerged as a rapidly advancing therapeutic strategy (Table 1). The majority of these molecules target FTO and METTL3.

The first selective METTL3 inhibitors were developed as competitive antagonists of the methyl donor SAM-binding site [71]. Based on this mechanism, a novel METTL3 inhibitor, STM2457, was identified. It binds with high efficiency to the METTL3-METTL14 complex and has been shown to impede the migration and self-renewal capacity of cancer stem cells in cellular and murine models of myeloid leukemia [72]. Moreover, STM2457 suppresses the tumor cell proliferation and growth in HCC cell lines and xenograft models (Figure 5B), while also reducing metastasis and enhancing sensitivity to chemotherapy in TNBC [73,74]. A derivative of this compound, STC-15, has become the first and currently the only small-molecule inhibitor based on METTL3 binding to be included in phase Ib/II clinical trials for the treatment of solid malignancies (NCT0558411).

Among eraser inhibitors, FTO-targeted compounds are studied the most. These include several small molecules, such as FTO inhibitors FB23 and FB23-2, built on a previously studied inhibitor, meclofenamic acid (MA) [75]. Both FB23 and FB23-2 suppress leukemia progression by selectively blocking FTO and restoring the m^6^A profile of *MYC*, *CEBPA*, *RARA*, and *ASB2* genes (Figure 5A) [76]. Subsequently, novel compounds CS1 and CS2 were identified as more effective inhibitors, binding and occupying FTO’s catalytic pocket [77]. CS1 and CS2 demonstrated suppression of leukemia proliferative activity and immune evasion, cancer cell viability, and leukemia stem cell renewal via *MYC*, *CEBPA*, and *RARA* axis [77]. Recent developments to regulate ALKBH5 demethylase activity include the covalent small-molecule inhibitor TD19, which disrupts the enzyme’s binding to target transcripts and reduces viability in acute myeloid leukemia and glioblastoma multiforme cell lines [78].

Small-molecule inhibitors targeting imbalanced m^6^A regulators (e.g., FTO, METTL3, and ALKBH5) show promising preclinical potential by destabilizing oncogenes (e.g., *MYC*/*CEBPA*) and overcoming therapy resistance. Key challenges include limited tumor specificity, poor bioavailability, and understanding of the functions and down-regulated pathways of the primary proteins, potential toxicity, and ineffective pharmacokinetic parameters [79]. Future studies require highly specific agents, advanced delivery systems, and combinative therapy.

**Table 1 molecules-30-04091-t001:** Small-molecule inhibitors targeting m^6^A regulators in cancer.

Target	Name of Inhibitor	Mechanism	Type of Cancer	Effects	Reference
METTL3/14	STM2457	Competitive inhibitor	TNBC	Decrease in cancer metastasis on the xenograft model in vivo; ↑ drug sensitivity;	[74]
			acute myeloid leukemia (AML)	Decline of proliferation, cell cycle arrest, and apoptosis induction in MOLM-13 cell line and primary mouse AML cells; ↓ protein levels of SP1 and BRD4.	[72]
	UZH1a	Competitive inhibitor	AML and osteosarcoma	Dose-dependent inhibition of METTL3 mRNA expression in the model of AML (MOLM-13 line) and osteosarcoma (U2OS line) cells.	[80]
	UZH2	Competitive inhibitor	AML	Inhibition of m^6^A demethylase activity resulting in global m^6^A hypomethylation and reduced viability in FLT3-ITD-mutated MOLM-13 AML cells.	[81]
	Quercetin	Competitive inhibitor	pancreatic cancer and HCC	Dose-dependent reduction in proliferation and viability of PaCa-2 and Huh7 tumor cell lines via lowering of mRNA METTL3 level. Its mechanism is associated with PI3K/Akt/mTOR, Wnt/b-catenin, and MAPK/ERK1/2 pathways.	[82]
	CDIBA	Allosteric inhibitor	AML	Highly selective inhibition of the METTL3/METTL14 complex and demonstration of anti-leukemia activity in AML cell lines including MOLM-13, MOLM-14, HL60, etc.	[83]
FTO ALKBH2 ALKBH3	Rhein	Competitive inhibitor	AML, HCC, pancreatic cancer	Regulation of the tumor cell cycle. Inhibition of AML cell proliferation and migration, also as apoptosis induction in cell models including THP1, HL60, MV4-11, etc. ↓ phosphorylated AKT and phosphorylated mTOR molecules. ↑ drug sensitivity.	[84,85]
FTO	MA (Meclofenamic acid)	Competitive inhibitor	NSCLC	The anti-proliferative effects on the normal PC9 and H292 and gefitinib-resistant PC9/GR and H292/GR NSCLC. ↑ drug sensitivity. ↓ expression level of *BCRP* and *MRP7*.	[86]
	FB23-2	Competitive inhibitor	AML	The anti-proliferative effects by the reducing the proliferation of NB4 and MONOMAC6 AML cells. Apoptosis initiation. ↓ expression levels of *MYC* and *CEBPA*.	[76]
	Dac51	Competitive inhibitor	Melanoma and LUAD	Reducing the cells and tumors in vitro and in vivo B16-OVA and MC38 models. ↓ *Jun*, *Cebpb*, and *Junb* mRNA and protein level. ↓ glycolytic metabolism.	[87]
	FTO-04	Competitive inhibitor	Glioblastoma	The impairment of glioblastoma stem cells (GSCs) self-renewal. Inhibition of GSCs’ neutrospheres formation.	[88]
	CS1/CS2	Competitive inhibitor	AML	Inhibition of cell proliferation/viability potential, increased apoptosis in AML cell lines (MONOMAC6, NOMO-1, and U937) and xenotransplantation models. The impairment of AML stem cells (LSCs) self-renewal. ↓ expression levels of *MYC*, *CEBPA*, and *RARA*.	[77]
	18097	Competitive inhibitor	Melanoma BC	Decreasing in proliferation, invasion, migration, and EMT of MDA-MB-231 and A375 cancer cells in vitro and BC tumor size in vivo. ↓ mRNA stability of *PPARG*, *CEBPA*, and *CEBPB*. ↑ mRNA stability of *SOCS1*. ↑ drug sensitivity.	[89]
ALKBH5	ALK-04	Competitive inhibitor	Melanoma	Decreasing in melanoma tumor size in vivo. ↑ drug sensitivity to GVAX/anti–PD-1.	[54]
	MV1035	Competitive inhibitor	Glioblastoma	Dose-dependent reduction in viability (A549 LCCs), inhibition of migration (U87-MG glioblastoma and H460 LCCs), and suppression of invasiveness in glioblastoma models. ↓ CD73 protein expression.	[90]
IGF2BP1	BTYNB	Allosteric inhibitor	Ovarian and melanoma cancers	Suppression of IMP1-positive IGROV-1, SK-MEL2 cells proliferation. ↓ *c-MYC* mRNA and protein levels.	[91]
IGF2BP2	CWI1-2	Competitive inhibitor	AML	Dose-dependent induction of apoptosis, mitochondrial dysfunction, and suppression of clonogenic potential in AML models with elevated IGF2BP2 expression.	[87]
YTHDF2	DC-Y13-27	-	Myeloid-derived suppressor cells (MDSCs)	Enhancing the antitumor effects of radiotherapy and radio-immunotherapy combinations and suppression of distant metastasis.	[92]

The up arrow indicates an increase, the down arrow indicates a decrease in protein or gene expression.

### 6.2. Antisense Oligonucleotides for Suppressing m^6^A-Related Factors

Beyond small-molecule inhibitors, antisense oligonucleotides (ASOs) offer another strategy to disrupt oncogenic m^6^A signaling through direct targeting of non-coding RNA, in particular, lncRNAs and miRNAs that function as critical regulators of writers, erasers, and readers within this epitranscriptomic system using RNA interference (RNAi) mechanism [93] (Table 2). Recent studies revealed a great variety of lncRNA (e.g., *NEAT1*, *MALAT1*, *CCAT2*, and *FOXM1*) and miRNA (e.g., miR-3662, miR-23b-3p, miRNA328, and miR-429) [3,94,95,96,97,98] that promote cell proliferation, angiogenesis, metastasis, and chemo- and immune-resistance in different types of cancers, such as lung adenocarcinoma (LUAD), lung squamous cell carcinoma (LUSC), BC, HCC, NSCLC, glioma, and acute myeloid leukemia.

Thus, antisense oligonucleotides were designed to target the lncRNA *PCAT6*, which induces cell proliferation and migration by forming a stability complex with the IGF2BP2 reader protein in prostate cancer (Figure 5C) [99]. The obtained results confirmed that ASO-mediated knockdown of *PCAT6* reduces bone metastasis and tumor growth in vivo [99]. Furthermore, ASO-mediated inhibition of the small nucleolar non-coding RNA *SNORD9* prevents its interaction with METTL3, downregulates NFYA mRNA expression levels, and, consequently, decreases the expression of its target genes (CCND1, CDK4, and VEGFA), suppressing ovarian cancer progression [100]. Additionally, antisense oligonucleotides targeting the lncRNA *STEAP3-AS1*, by disrupting its binding to YTHDF2, negatively impact proliferation, migration, and invasion in colorectal cancer (Figure 5D) [101]. Moreover, ASO-mediated inhibition of the circular RNA *circPLPP4* overcomes cisplatin chemoresistance in ovarian cancer by modulating the METTL3/PI3K-AKT signaling pathway [102].

Generally, these studies demonstrate the significant therapeutic potential of ASOs in targeting various m^6^A-associated non-coding RNAs to effectively and selectively suppress oncogenic processes across multiple cancer types. However, this approach remains under active development, with several fundamental difficulties requiring resolution: optimizing efficient targeted delivery systems; enhancing the safety and tolerability characteristics of ASOs; and improving binding affinity and specificity for target ncRNAs [103,104].

**Table 2 molecules-30-04091-t002:** ASO-based therapies targeting m^6^A-related pathways in cancer.

Target	Type of Cancer	Effects	Reference
*SNORD9*	Ovarian cancer	Suppression of METTL3 and IGF2BP2 mRNA and protein levels. Reduced cell proliferation and migration for ovarian cancer models.	[100]
*METTL3*	Prostate cancer	mRNA and protein reductions in METTL3 and the ERK signaling axis in vitro and in vivo. ↑ drug sensitivity.	[105]
*PCAT6*	Prostate cancer	↓ brain metastasis and tumor growth in vivo via *PCAT6*/*IGF2BP2*/*IGF1R* pathway.	[99]
*METTL3*	Cholangiocarcinoma (CCA)	Inhibition of CCA cell proliferation, colony formation, and migration/invasion in vitro and prevents CCA development/progression in mice. Decreased the level of the Hippo-TAZ pathway.	[106]
*FTO-IT1*	Prostate cancer	Inhibition of tumor cell cycle proliferation and colony formation ability, and induced apoptotic cell death in vivo with p53 activation.	[107]
*LINC00839*	Glioblastoma	The ASP-associated knockdown significantly enhanced therapeutic sensitivity and apoptotic response while concurrently reducing clonogenic survival capacity via METTL3/YTHDF2 expression decline.	[108]

The up arrow indicates an increase, the down arrow indicates a decrease in protein or gene expression.

### 6.3. Utilizing CRISPR/Cas for Manipulating with m^6^A Sites

Compared with selective inhibitors and antisense oligonucleotide strategies, point epigenome editing mechanisms have emerged using nuclease-deficient CRISPR-dCas9 (dead Cas9) fused to functional domains of m^6^A machinery. This approach enables site-specific m^6^A modification: fusions with catalytic domains of methyltransferase complex (METTL3-METTL14) modulate m^6^A-guided writers M3M14-dCas9, while fusions with the full-length of FTO or ALKBH5 create m^6^A-guided erasers FTO-dCas9 and ALKBH5-dCas9 [109]. Thus, the m^6^A-editing system enables targeted installation or removal of m^6^A marks at specified genomic loci without deletion of regulatory elements from signaling pathways or altering global cellular m^6^A levels [110]. Subsequent m^6^A-editing platforms were optimized by replacing dCas9 with dCas13, which demonstrated significantly reduced off-target methylation rates while expanding target range through independence from protospacer flanking sequence (PFS) limitations [111,112]. Moreover, dCas13 enables nuclear RNA manipulation, overcoming dCas9′s cytoplasmic restrictions, and possesses a compact protein size that enhances delivery efficiency for therapeutic applications [111].

Among the tools developed for targeted editing of epigenetic marks in mammalian cells is the targeted RNA methylation (TRM) system (Figure 5C). This system was engineered by fusing catalytically dead Cas13 (dCas13) with two distinct methyltransferase configurations. One version links dCas13 to METTL3, lacking its zinc finger RNA-binding motifs and comprising a nuclear localization signal (NLS), termed dCas13-M3nls. Another version fuses dCas13 to the METTL3-METTL14 heterodimer complex, comprising a nuclear export signal (NES), named dCas13-M3M14nes [113]. Wilson et al. confirmed the programmed methylation of both structures in *FOXM1* and *SOX2* genes; however, dCas13-M3M14nes demonstrated a higher off-target effect without significantly increasing target activity.

Complementing these methylation tools, systems for targeted RNA demethylation at unique sites have also been developed. These generally comprise several approaches: the dCas13b-ALKBH5nes fusion (named dm^6^ACRISPR), which was one of the first systems to program demethylase manipulation in vivo; dCas13Rx directly fused to ALKBH5 for the possibility to be packed into AAV or lentiviral vectors and multi-site activity; and the TRADES system, which leverages dCas13b-GCN4 coupled with single-chain variable fragment (scFv) fusions to either FTO or ALKBH5 for a wider targeted demethylation window [114,115,116]. Furthermore, Rauch et al. developed dCas13b-m^6^A reader tools by fusing catalytically inactive PspCas13b with the m^6^A reader proteins YTHDF1 and YTHDF2 [14]. These chimeric complexes enable targeted effects on translational enhancement (via YTHDF1) or RNA decay (via YTHDF2) of specific endogenous transcripts. The system allows for the investigation of dynamic RNA regulation in liver cells by switching between the YTHDF1 and YTHDF2 effector domains. A particularly interesting tool is the optogenetic system PAMEK (programmable m^6^A editor with light). PAMEK uses blue light to control m^6^A modulation [108]. It can specifically install m^6^A marks using the METTL3-METTL14 complex or remove them using full-length FTO protein on targeted RNA transcripts such as *MALAT1*, *TPT1*, and *ACTB*. The high spatiotemporal resolution of this system is achieved by application of light-dependent protein interaction between the dCas13-fused CIBN (CIB1) domain and the effector protein CRY2PHR, which is covalently conjugated to the respective m^6^A-modifying enzyme [117]. This platform enables light-inducible gene expression changes via m^6^A level alterations in selected molecules, facilitating analysis of epigenetic tag influences on specific genes and identification of prospective therapeutic targets.

These targeted editing tools demonstrate functional impacts in cancer models. Specifically, lentiviral delivery of dCas13Rx-METTL3 and dCas13Rx-ALKBH5 enables precise m^6^A enrichment on *FOXM1* mRNA and m^6^A depletion on *MYC* mRNA, respectively (Figure 5E). This targeted epitranscriptome editing reduces the proliferation of glioblastoma cells [118]. Similarly, the dm^6^ACRISPR system reduces m^6^A levels on *EGFR* and *MYC* mRNA, leading to decreased transcript expression and inhibition of cellular activity in HeLa cells (Figure 5F) [114].

Generally, the development of CRISPR-based systems based on Cas9 and Cas13 proteins for targeted m^6^A modification, including both reprogrammable site-specific methylation and demethylation, represents a significant advance in understanding and searching for clinically relevant mechanisms of tumor cell adaptation by influencing the RNA’s fate. Despite the promise of epitranscriptome engineering, several significant challenges complicate its practical application. Key limitations that must be addressed include the potential for off-target effects, powerless editing efficiency, constraints in delivery efficiency to target cells or tissues, and the need for enhanced specificity of the editing tools [119]. Overcoming these hurdles is critical for advancing the translational potential of RNA modification technologies.

Alternative approaches may involve sustained knockdown of m^6^A-related genes using dCas9-based transcription inactivation systems, such as CRISPRoff [120]. These systems function by recruiting epigenetic silencers (e.g., KRAB, DNMT3A) to gene regulatory elements—promoters and enhancers—thereby blocking transcription. Unlike ASOs, which transiently suppress gene expression without establishing stable epigenetic modifications, CRISPR/Cas-mediated transcriptional inactivation induces long-term gene silencing through the formation of an “epigenetic memory” at the target locus [121]. Similarly efficient but prone to on-target genetic aberrations, gene-editing tools—such as nucleases and base-editing systems—can irreversibly disrupt gene function by mutating the start codon.

Future advancements in epitranscriptome engineering will critically focus on three key areas: the engineering of Cas proteins (including optimization of size and functionality), the development of advanced delivery technologies such as adeno-associated viruses (AAVs) or nanoparticle-based systems, and the acquisition of deeper insights into target RNA molecules and their associated signaling pathways in pathological and physiological contexts [122]. Progress in these areas is essential for unlocking the full clinical potential of RNA modification tools in numerous diseases, including various types of cancer.

## 7. Concluding Remarks

m^6^A RNA modification is one of the fundamental processes in normal cellular physiology, such as embryonic development [123], gametogenesis, immunoregulation [124], and carcinogenesis [125]. Research consistently demonstrates that impaired coordination within the m^6^A modification system, particularly the crosstalk between writers, erasers, and readers, is associated with adverse outcomes and drug resistance across various malignancies, such as lung cancer, breast cancer, melanoma, and hepatocellular carcinoma [112,126]. As m^6^A modification occurs on approximately one-third of all mRNAs, such dysregulation plays a substantial role in driving disease-promoting cellular phenotypes [127]. For these reasons, proteins within the m^6^A machinery are promising candidates for investigation as targets for cancer screening, early diagnostic approaches, and potential therapeutics.

By analyzing the role of the modification of mRNA and ncRNA in cancer progression, researchers confirmed strong participation, mostly regarding METTL3/METTL14 complex, FTO, and ALKBH5, in tumor cell proliferation, invasion, migration, angiogenesis, metastasis, and drug resistance development [128]. However, their roles are context-dependent. Reduced activity of METTL14 and ZC3H13 methyltransferases correlates with lower survival and acts as a tumor-suppressive in breast cancer [47]. Conversely, elevated expression of METTL3 was determined in AML, bladder cancer, gastric cancer, and liver cancer, where it had been suggested to function as an oncogene, suggesting critical mechanisms for tumor progression [129,130]. The demethylase ALKBH5 with m^6^A binding proteins YTHDF1/2/3, concurrently, played a suppression role in the regulation of NSCLC tumor growth and metastasis [131], while FTO inhibits tumorigenesis in prostate cancer [132]. These findings highlight how m^6^A regulators, by modifying the epitranscriptomic profile, promote tissue-specific control over oncogenesis in cancers. Despite the considerable evidence for m^6^A proteins in oncogenesis, their clinical utilization as therapeutic targets or biomarkers encounters limitations.

A major limitation in clinically targeting the m^6^A machinery is its functional duality, even within the same cancer type. For example, in breast cancer, METTL3-mediated methylation exhibits contrary roles: it can function as a tumor suppressor [42] in some contexts while acting as an oncogenic encourager [67] by maintaining tumor cell stemness in others. Similarly, the demethylase ALKBH5 demonstrates opposing functions in HCC: it inhibits malignancy and progression [61] in certain cases and promotes oncogenesis [133] by stabilizing pro-oncogenic mRNA in HBV-related HCC. The development of reliable therapeutic strategies is a significant challenge due to the context-dependent duality that requires an in-depth study of the molecular mechanisms of RNA methylation.

Context-dependent duality poses fundamental challenges, and the development of therapeutic strategies that target m^6^A regulatory proteins is still in its initial stage. While promising in silico discoveries have been made, including small-molecule inhibitors designed against specific structural features of key regulators like METTL3 (e.g., STM2457) and FTO (e.g., CS1/CS2), as well as antisense oligonucleotides (Figure 6A), robust validation in physiologically relevant models, difficulties in delivery efficiencies, lack of confirmed clinical biomarkers, and insufficient data from wide preclinical trials continue to slow the advancement of these strategies toward clinical trials [134]. Therefore, there are still a lot of unanswered questions about whether these potential therapies can achieve the required levels of target specificity and effective inhibition needed for safe and effective clinical application.

To overcome the significant challenges, targeted epigenetic editing via CRISPR/Cas9 (Cas13) systems represents a highly promising strategy for m^6^A methylation profile regulation (Figure 6A). This approach offers a potential solution to overcome the key limitation of off-target effects and lack of specificity, which are deficiencies identified in the previously mentioned therapeutic strategies. Primarily, CRISPR/Cas enables the precise addition or removal of m^6^A marks at specific sites within target transcripts [111]. This site-specific strategy avoids systemic disruption of signaling pathways and RNA metabolism, preventing disturbance of the conserved writer–eraser–reader connection and mitigating risks arising from their functional duality and close interactions within the molecular network.

Despite its prospective promises, the clinical application of CRISPR/Cas-based m^6^A editing systems faces significant limitations (Figure 6B). These include delivery limitations, low editing efficiency, off-target opportunity, technical challenges of multiplex editing, and toxicity effects [111,119]. As a result, overcoming current restrictions requires directed efforts on optimizing vector constructs to reduce fusion protein size, enhancing fusion protein design for improved specificity and catalytic activity, optimizing gRNA sequences to maximize on-target binding precision, and developing advanced tissue-specific delivery systems for efficient in vivo targeting. Finally, resolving these challenges and uncovering the context-specific fundamental principles of RNA methylation will provide a lot of opportunities for the m^6^A system in clinical applications. It could be determined as a safe and effective therapeutic agent, used either as a self-treatment or as part of combined therapy against aggressive malignant tumors.

## Figures and Tables

**Figure 1 molecules-30-04091-f001:**
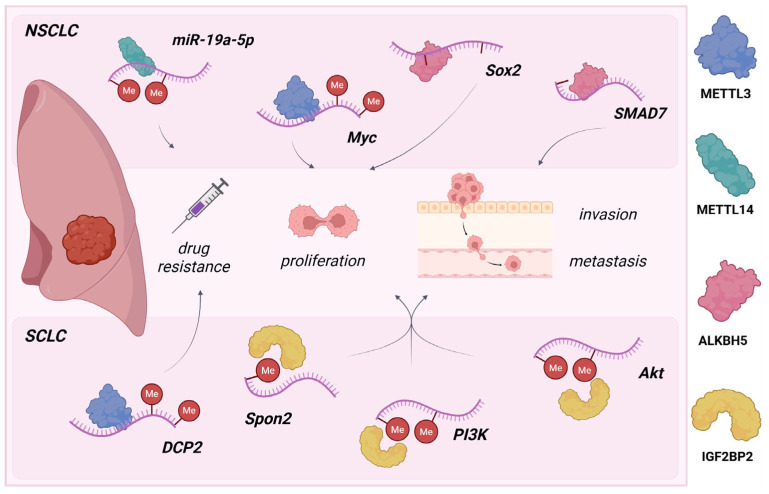
The m^6^A regulatory network and its oncogenic roles in lung cancer progression. Overactivity of writers (METTL3/METTL14), erasers (ALKBH5), and readers (IGF2BP2) drives the oncogenic behaviors, particularly the proliferation, invasion, metastasis, and drug resistance of tumor cells in lung cancer. Created in www.biorender.com (accessed on 6 October 2025).

**Figure 2 molecules-30-04091-f002:**
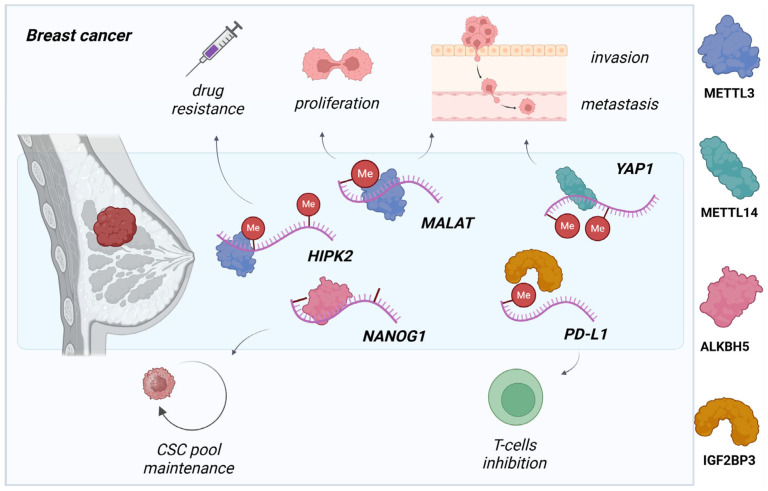
The m^6^A epitranscriptomic machinery in breast cancer progression. Upregulation of METTL3/METTL14, ALKBH5, and IGF2BP3 enhances metastatic potential and suppresses T cell activity via specific transcript regulation in breast cancer. Created in www.biorender.com (accessed on 6 October 2025).

**Figure 3 molecules-30-04091-f003:**
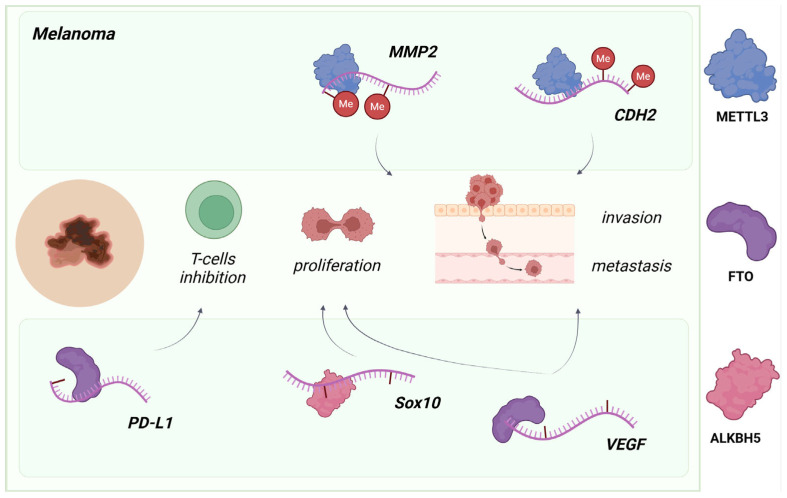
The active role of m^6^A regulatory proteins in melanoma. The imbalance between writer (METTL3) and eraser (FTO/ALKBH5) activities encourages melanoma progression by proliferation, invasion, metastasis, and/or transformation of immune microenvironment. Created in www.biorender.com (accessed on 6 October 2025).

**Figure 4 molecules-30-04091-f004:**
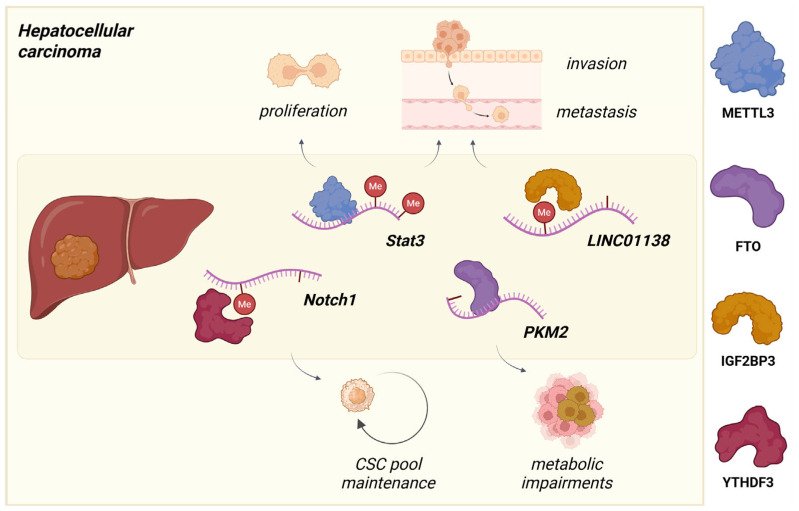
Oncogenic roles of m^6^A epitranscriptomic system in hepatocellular carcinoma progression. Dysregulation of METTL3, FTO, IGF2BP3, and YTHDF3 maintains cancer stem cell pool and promotes glycolytic metabolism in hepatocellular carcinoma cells. Created in www.biorender.com (accessed on 6 October 2025).

**Figure 5 molecules-30-04091-f005:**
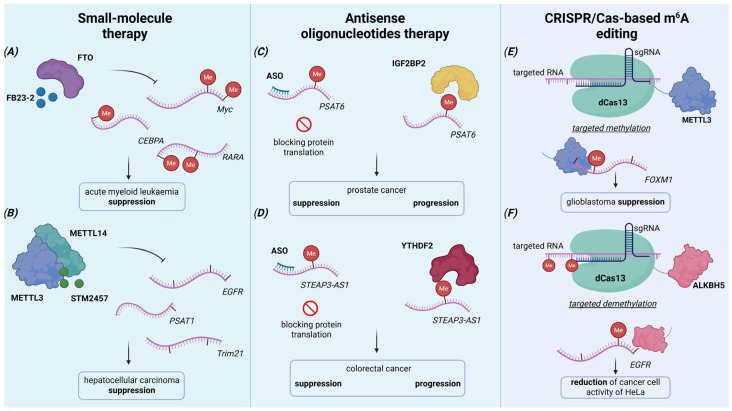
Schematic representation of strategies to target m^6^A RNA methylation for cancer therapy. (**A**,**B**) Blocking writers (METTL3/14) or erasers (FTO, ALKBH5) with *small inhibitors* alters the m^6^A landscape, destabilizing the downregulated pool of oncogenic transcripts (*MYC*, *EGFR*, and *CEBPA*). (**C**,**D**) Therapy with *small interfering molecules* silences target RNA and its downregulated signaling axis. (**E**,**F**) *CRISPR/Cas*-directed m^6^A *editing* allows targeted methylation or demethylation of individual RNA transcripts, modulating their processing without affecting the overall m^6^A distribution. Combined approaches may inhibit cancer progression by reprogramming the m^6^A epitranscriptome to suppress oncogenic signaling, metastasis, and therapy resistance. Created in www.biorender.com (accessed on 6 October 2025).

**Figure 6 molecules-30-04091-f006:**
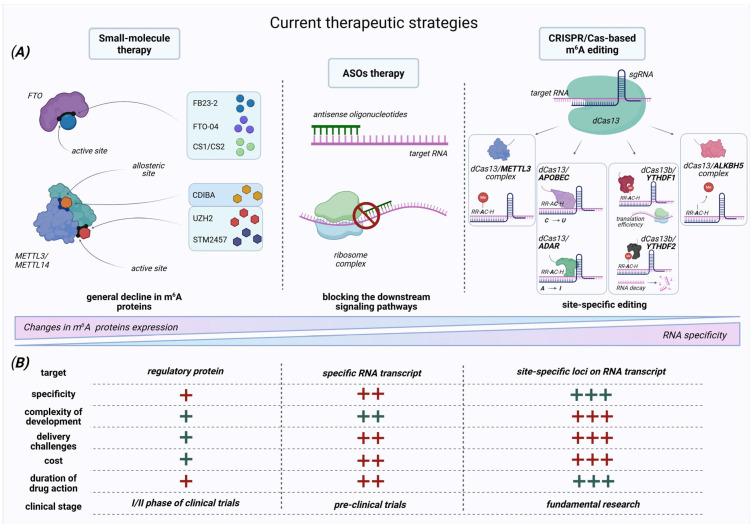
Current approaches to modifying the m^6^A epitranscriptome profile. (**A**) This schematic summarizes three leading strategies for targeted manipulation of the m^6^A epitranscriptome in cancer therapy: (1) *small-molecule inhibitors* that selectively target m^6^A regulatory proteins (methyltransferases and demethylases) through competitive or allosteric binding to modulate their activity; (2) *antisense oligonucleotides (ASOs)* that disrupt the interaction between m^6^A regulatory proteins and their target RNAs; and (3) *CRISPR/Cas-based editing* systems that enable site-specific m^6^A modifications through targeted methylation/demethylation via METTL3- or ALKBH5-binding complex, introduction of mutations in RNA sequences using ADAR or APOBEC proteins, or regulation of transcript stability and translation by engaging reader proteins such as YTHDF1 and YTHDF2. (**B**) A comparative analysis of current therapeutic strategies indicates that the use of *small-molecule inhibitors* represents the most cost-efficient and rapidly advancing approach. One such compound, STC-15, is currently in the early stages of clinical investigation. However, the primary limitation of this strategy is its lack of specificity. Since these inhibitors target the activity of key regulatory proteins, they can induce widespread alterations in the methylation profile of transcripts that are critical for normal cellular viability and function, forming a significant risk of on-target toxicity. The utilization of *ASOs* is more complex and expensive; however, they operate with higher precision, allowing for the targeted suppression of individual oncogenic transcripts. This localized action significantly reduces, though does not completely eliminate, the risk of disrupting essential downstream regulatory pathways. CRISPR/Cas m^6^A editing tools are the most expensive and complicated among others. This system allows for site-specific methylation changes within the transcript to minimize undesirable phenomena and provide the longest duration of action as a therapeutic drug. The green color (+) means advantages, the red color (+) means disadvantages. The number of pluses indicates the level of difficulty. Created in www.biorender.com (accessed on 6 October 2025).

## Data Availability

No new data were created or analyzed in this study.

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
