# Peer review of "m6A RNA Modification: Technologies Behind Future Anti-Cancer Therapy"

_molecules, 2025, doi:10.3390/molecules30204091_

Round 1
Reviewer 1 Report
Comments and Suggestions for Authors
The manuscript is structured clearly, scientifically sound, and effectively communicates the biological and therapeutic significance of m6A modifications in a logical and organized manner. The images are of exceptional quality and serve as a creative summary of each section.The introduction clearly explains the significance of m6A marks and their dual function in cancer biology, underscoring the relevance of this review.It is important to point out that the emphasis on the equilibrium between methylation and demethylation processes, as well as the concept of "writers, erasers, and readers," illustrates a comprehensive comprehension of the field and describes the intricacy of modifications of RNA in epigenetics.
Nevertheless, there are a few critical areas in which enhancements can be implemented.
- The overwhelming majority of the works referenced in this review are derived from in vitro experiments. A section on the prospective use of m6A modifiers in translational medicine, as well as the limitations and prospects of clinical trials targeting m6A modifiers in cancer, should be included by the authors.
- The authors do not specify the image processor they employed to generate the images (e.g., BioRender); I believe that the copyright for each image should be included at the end of the figure legend
Author Response
We sincerely thank the Reviewers for their thorough evaluation of our manuscript and their valuable comments, which have significantly improved its quality. Our point-by-point responses are provided below.
Comment 1. The overwhelming majority of the works referenced in this review are derived from in vitro experiments. A section on the prospective use of m6A modifiers in translational medicine, as well as the limitations and prospects of clinical trials targeting m6A modifiers in cancer, should be included by the authors.
Response 1. We sincerely thank the Reviewer for this insightful comment and for highlighting the importance of the clinical translation perspective regarding m6A modifiers. We fully agree that this is a critical area for the field.
In revising the manuscript, we carefully considered the suggestion to add a dedicated section. However, for the following reasons, we have chosen to integrate this crucial discussion throughout the existing narrative structure rather than as a separate chapter: 1) the field remains predominantly preclinical, with limited clinical trial data available, and a separate section might overstate current clinical applicability; 2) we wish to maintain alignment with the journal's recommended scope and length.
To directly address the Reviewer's valuable point, we have now integrated discussions of clinical translation and trial limitations throughout the manuscript in the following sections:
1) We added the current clinical application of m6A functional proteins in diagnosis and prognosis of cancer development in lines 122-133:
“In current clinical practice, the dynamic profile of m6A RNA modification, along with the expression levels of its regulatory proteins, shows promise as a prognostic biomarker and is directly linked to outcomes in various cancer types, e.g., elevated expression of YTHDF1 and IGF2BP2 correlates with worse overall survival (OS) in HCC [26]. However, the clinical translation of m6A regulators as therapeutic targets faces several challenges, including the complexity of developing targeted therapies due to the widespread nature of RNA modifications, their dual roles in essential cellular processes, potential toxic effects, and delivery difficulties. Despite these challenges, several therapeutic approaches are under investigation. Small-molecule inhibitors and antisense oligonucleotides are currently in early stages of preclinical and clinical development. In contrast, CRISPR/Cas-based technology, while promising, remains at the fundamental research stage due to challenges in reproducibility.”
2) We underlined the field of m6A RNA modification is still developing from fundamental research to clinical application due to existing limitations, and active clinical trials are still in the early stages.
- in lines 345-347: “A derivative of this compound, STC-15, has become the first and currently the only small-molecule inhibitor based on METTL3 binding to be included in phase Ib/II clinical trials for the treatment of solid malignancies (NCT0558411)”.
- in lines 533-542: “Context-dependent duality poses fundamental challenges, and the development of therapeutic strategies that target m6A regulatory proteins is still in its initial stage. While promising in silico discoveries have been made, including small-molecule inhibitors designed against specific structural features of key regulators like METTL3 (e.g., STM2457) and FTO (e.g., CS1/CS2), as well as antisense oligonucleotides, robust validation in physiologically relevant models, difficulties in delivery efficiencies, lack of confirmed clinical biomarkers and insufficient data from wide preclinical trials continue to slow the advancement of these strategies toward clinical trials [125]. Therefore, there are still a lot of unanswered questions about whether these potential therapies can achieve the required levels of target specificity and effective inhibition needed for safe and effective clinical application”
Comment 2. The authors do not specify the image processor they employed to generate the images (e.g., BioRender); I believe that the copyright for each image should be included at the end of the figure legend.
Response 2. Thank you for pointing this out. We have added the suggested information to the manuscript at the end of each figure legend.
Reviewer 2 Report
Comments and Suggestions for Authors
This review explores research advances in targeting N6-methyladenosine (m6A) RNA modification for cancer treatment, highlighting its impact on tumor progression, metastasis, and drug resistance. It highlights that targeting m6A machinery via small-molecule inhibitors, antisense oligonucleotides, and CRISPR/Cas-based tools offers promising anti-cancer strategies, though challenges remain in specificity and efficacy. This review provides good coverage of m6A's biological significance and therapeutic potential, with well-organized content that enhances clarity and comprehension. I have only two minor comments:
- The phrase “m6A RNA editing” is confounding since “RNA editing” is another biological concept, consider replace it with “Editing m6A” or “Modulating m6A”.
- Figure 2, the “host immune response” in the figure legend is confounding since there is no pathogen infection here, simply “immune response” is enough, while “immune microenvironment” or “T cell suppression” would be a better choice.
Author Response
We sincerely thank the Reviewers for their thorough evaluation of our manuscript and their valuable comments, which have significantly improved its quality. Our point-by-point responses are provided below.
Comment 1. The phrase “m6A RNA editing” is confounding since “RNA editing” is another biological concept, consider replace it with “Editing m6A” or “Modulating m6A”.
Response 1. We sincerely thank the Reviewer for this accurate observation. To avoid confusion with the canonical concept of RNA editing, we have replaced all instances of "m6A RNA editing" with the recommended terms, "editing m6A profile" and "modulating m6A level", as well as other context-appropriate alternatives.
Comment 2. Figure 2, the “host immune response” in the figure legend is confounding since there is no pathogen infection here, simply “immune response” is enough, while “immune microenvironment” or “T cell suppression” would be a better choice.
Response 2. Thank you for this suggestion. We have revised the text as suggested and replaced the previous phrasing with "suppression of T-cell activity" or “transformation of immune microenvironment” throughout the relevant sections.
Reviewer 3 Report
Comments and Suggestions for Authors
The authors of the manuscript "m6A RNA editing: technologies behind future anti-cancer therapy" have done a commendable job providing a comprehensive role of N6-methyladenosine (m6A) RNA modification system and its complex, context-dependent roles in multiple cancers. The authors have highlighted recent advances in targeting the m6A machinery, including small molecule inhibitors, antisense oligonucleotides and CRISPR/Cas-based editing tools. The manuscript is well written with adequate examples and representations in the form of figures and tables.
There is however, a minor concern that I would like the authors to respond to.
The authors should discuss about more multi omics techniques and single cell techniques that help cover a more global response to such m6A edits. Furthermore, in recent times we have robust and extremely informative microscopy studies that complement these edits. I strongly believe the authors should probably come up with a section and/or a figure where they also summarize how post m6A edits either with small molecule inhibitors, ASO and CRISPR/Cas based editing tools can benefit from a comparative study of the readouts with respect to the ease, cost and duration of final results.
Author Response
We sincerely thank the Reviewers for their thorough evaluation of our manuscript and their valuable comments, which have significantly improved its quality. Our point-by-point responses are provided below.
Comment 1: The authors of the manuscript "m6A RNA editing: technologies behind future anti-cancer therapy" have done a commendable job providing a comprehensive role of N6-methyladenosine (m6A) RNA modification system and its complex, context-dependent roles in multiple cancers. The authors have highlighted recent advances in targeting the m6A machinery, including small molecule inhibitors, antisense oligonucleotides and CRISPR/Cas-based editing tools. The manuscript is well written with adequate examples and representations in the form of figures and tables.
There is however, a minor concern that I would like the authors to respond to.
The authors should discuss about more multi omics techniques and single cell techniques that help cover a more global response to such m6A edits. Furthermore, in recent times we have robust and extremely informative microscopy studies that complement these edits.
Response 1: We are grateful to the Reviewer for this insightful comment. In response, we have added a comprehensive discussion of the methodologies available for m6A detection and visualization, encompassing both in vitro and in situ approaches in Introduction part in lines 79-109. We believe this new content addresses the Reviewer's point and significantly enhances the completeness and scope of our manuscript.
Comment 2: I strongly believe the authors should probably come up with a section and/or a figure where they also summarize how post m6A edits either with small molecule inhibitors, ASO and CRISPR/Cas based editing tools can benefit from a comparative study of the readouts with respect to the ease, cost and duration of final results.
Response 2: We thank the Reviewer for the feedback. The manuscript has been updated with a revised figure (Figure 6), that systematically outlines the key benefits and limitations of each therapeutic approach. To enhance navigation and understanding, we have added numerical labels directly to the figure and referenced these labels in the corresponding sections of the main text.
Reviewer 4 Report
Comments and Suggestions for Authors
The paper summarizes the functional regulation roles of m6A in various cancer contexts and also introduces the three common strategies for targeting m6A regulation for cancer therapeutic development. Overall, the review is well written with only two minor edits necessary to complement the content.
1. In the introduction section, it is also worth mentioning that m6A has also been found on non-coding RNAs (generally, not only lncRNAs), especially in some species on chromatin, which are critical for m6A-dependent transcription regulation.
2. In the final section before the conclusion on dCas13 and dCas9, it will be helpful to also mention that the dCas13 system has also been incorporated with the reader proteins such as YTHDF1 and YTHDF2, which help establish a wider scope in applying the dCas system for targeting m6A pathways.
Author Response
We sincerely thank the Reviewers for their thorough evaluation of our manuscript and their valuable comments, which have significantly improved its quality. Our point-by-point responses are provided below.
Comment 1. In the introduction section, it is also worth mentioning that m6A has also been found on non-coding RNAs (generally, not only lncRNAs), especially in some species on chromatin, which are critical for m6A-dependent transcription regulation.
Response 1. We thank the Reviewer for this comment. The Introduction has been revised to highlight the crucial roles of non-coding RNAs, particularly microRNAs and circRNAs, as suggested.
Comments 2. In the final section before the conclusion on dCas13 and dCas9, it will be helpful to also mention that the dCas13 system has also been incorporated with the reader proteins such as YTHDF1 and YTHDF2, which help establish a wider scope in applying the dCas system for targeting m6A pathways.
Comments 2. We appreciate you pointing this out. We have added the suggested content about dCas13b-YTHDF1/YTHDF2 tools to your recommended part of manuscripts on lines 444-449.
Reviewer 5 Report
Comments and Suggestions for Authors
The manuscript I have in my possession, "m6A RNA editing: technologies behind future anti-cancer therapy" by Shpilyukova (molecules-3903355), is a review article intended to provide an overview of the current data on the molecules METTL3 and METTL14, which are involved in m6A RNA editing, with regard to selected tumor entities. Although this review article addresses a current research topic and is interestingly written, it requires extensive corrections before a decision can be made as to whether this work should be accepted.
Major comments:
- The title is misleading and does not fit the manuscript. The authors focus exclusively on the molecules METTL3 and METTL14, and this must be reflected in the title!
- The introduction is absolutely superficial with regard to the focus on METTL3 and METTL14. All (!) molecular processes of this m6A modification must be discussed in more detail so that readers can understand the following inhibitor strategies.
- Figure 2 may look quite cute at first glance, but it is extremely confusing. Some kind of numbering or meaningful ordering is necessary.
- The inhibitor strategies must be compared and weighed against each other. For cost reasons alone, CRISPR/CAS9 and the antisense oligos may soon lose their relevance. Therefore, the far too long section on CRISPR/CAS9 inhibition must be significantly shortened.
- A final, summarized schematic representation of all the key findings of the review is necessary.
Minor comments
- Avoid abbreviations in headings, e.g., BC
- Rephrase: "....In 2018, BC..." sounds more like the introduction to a history lesson on antiquity.
But overall an interesting work.
Author Response
We sincerely thank the Reviewers for their thorough evaluation of our manuscript and their valuable comments, which have significantly improved its quality. Our point-by-point responses are provided below.
Major comments:
The title is misleading and does not fit the manuscript. The authors focus exclusively on the molecules METTL3 and METTL14, and this must be reflected in the title!
The introduction is absolutely superficial with regard to the focus on METTL3 and METTL14. All (!) molecular processes of this m6A modification must be discussed in more detail so that readers can understand the following inhibitor strategies.
Response
Thank you for pointing this out. We agree that our manuscript provides substantial focus on methyltransferases, and we appreciate the opportunity to clarify its scope. As the Reviewer suggested, we have now expanded the discussion of reader proteins, including their roles in fundamental cellular processes in the Introduction. We have also added a section on the innovative CRISPR/dCas13 technology, which applies YTHDF1 and YTHDF2 to manipulate RNA fate.
Following a thorough revision and with the addition of this new content, we believe the original title accurately reflects the review's comprehensive nature, as we strive to cover all key aspects of m6A RNA modification, including the roles of writers, erasers, and readers.
Major comments
Figure 2 may look quite cute at first glance, but it is extremely confusing. Some kind of numbering or meaningful ordering is necessary.
Response
We thank the Reviewer for the insightful comment. In response, we have improved the figure by splitting it and integrating a numbered structure directly into the image and its legend. These new structures are now directly referenced in the main text, allowing for a clearer explanation and better emphasis of the critical details for each described cancer type in their respective chapters.
Major comments
The inhibitor strategies must be compared and weighed against each other. For cost reasons alone, CRISPR/CAS9 and the antisense oligos may soon lose their relevance. Therefore, the far too long section on CRISPR/CAS9 inhibition must be significantly shortened.
A final, summarized schematic representation of all the key findings of the review is necessary.
Response
We agree with the reviewer's comment and have modified the figure accordingly, which is now labeled as Figure 6. This revised figure summarizes the most critical data on current therapeutic approaches for m6A RNA modification. It includes a schematic overview of the mechanisms of action for small-molecule inhibitors, ASOs, and CRISPR/Cas systems, alongside a comparative analysis of their advantages and disadvantages. After careful consideration, we decided to maintain the section on CRISPR/Cas-based technologies because it is widely accepted as a transformative approach, even if not yet clinically mature. Its unique ability to manipulate the RNA epitranscriptome makes it an essential tool for future target discovery and therapy development.
Minor comments
Avoid abbreviations in headings, e.g., BC
Rephrase: "....In 2018, BC..." sounds more like the introduction to a history lesson on antiquity.
Response. We thank the Reviewer for their constructive comments. The manuscript has been revised accordingly to improve its scientific quality and accuracy.
Reviewer 6 Report
Comments and Suggestions for Authors
Dear Authors,
the present manuscript describes an interesting topic.
English is OK but could be improved, expecially some sentences and the figures The comments below suggest some changes.
Major issues
- lines 69-74 should be rephrased, sentences are not clear.
- line 80: "most prevalent human cancers"
- Figure 2: although the figure has been prepared with Biorender and is of high quality, the presentation is not satisfactory. The content is very dense and it is not clear to understand. I suggest to use larger panels, ordered vertically, not side by side. One tumor at a time. In each paragraph specific for each tumor the panel of the figure should be indicated.
- Figure 3: this figure presents mechanisms that are not obvious. The effect of suppression or activation should be represented differently. Not in a box. I would rather use arrows. Overall, Figure 3 is not clear, even in the Crispr/CAS panel, the mechanism of inhibition-modulation is not understandable. Panels MUST be specified and indicated in the legend and text. The overall organization of the figure must be improved.
Minor issues
- the title should be changed. "... behind future anti-cancer therapies" does not sound correct.
- Introduction. Speaking of cancer, the recent article concerning m6A in cancer should be cited: doi: 10.3389/fonc.2023.1063636.
- line 233: "... FTO was determines..." must be changed.
- line 271: "the most amounts of these molecule target on..." must be changed.
- Table: please, use also vertical lines, in addition to horizontal.
- Paragraph 7: I would write "Concluding remarks".
I suggest a professional English proofreading.
Author Response
We sincerely thank the Reviewers for their thorough evaluation of our manuscript and their valuable comments, which have significantly improved its quality. Our point-by-point responses are provided below.
Comment 1: lines 69-74 should be rephrased, sentences are not clear.
line 80: "most prevalent human cancers"
Response. We thank the Reviewer for their valuable feedback. We have revised the manuscript accordingly to enhance its scientific clarity.
Comment 2: Figure 2: although the figure has been prepared with Biorender and is of high quality, the presentation is not satisfactory. The content is very dense and it is not clear to understand. I suggest to use larger panels, ordered vertically, not side by side. One tumor at a time. In each paragraph specific for each tumor the panel of the figure should be indicated.
Response 2: We are grateful for this helpful comment. In response, we have split the figure into four distinct parts and incorporated numerical labels into each panel and their corresponding legends to facilitate better cross-referencing in the text. To prevent any potential misunderstanding of the mechanisms depicted for each cancer type and to enhance the manuscript's clarity and logical flow, we have inserted each relevant figure section into its corresponding chapter.
Comment 3: Figure 3: this figure presents mechanisms that are not obvious. The effect of suppression or activation should be represented differently. Not in a box. I would rather use arrows. Overall, Figure 3 is not clear, even in the Crispr/CAS panel, the mechanism of inhibition-modulation is not understandable. Panels MUST be specified and indicated in the legend and text. The overall organization of the figure must be improved.
Response 3: We thank the Reviewer for their constructive suggestion. The figure has been modified to enhance visual clarity and improve the explanation of the CRISPR/Cas-based m6A editing system's mechanism. In accordance with the comment, we have added numerical labels to the individual parts of the figure (now Figure 6) and have referenced these labels in the relevant sections of the manuscript to provide a clearer guide for the reader.
Minor issues
- the title should be changed. "... behind future anti-cancer therapies" does not sound correct.
- Introduction. Speaking of cancer, the recent article concerning m6A in cancer should be cited: doi: 10.3389/fonc.2023.1063636.
- line 233: "... FTO was determines..." must be changed.
- line 271: "the most amounts of these molecule target on..." must be changed.
- Table: please, use also vertical lines, in addition to horizontal.
- Paragraph 7: I would write "Concluding remarks".
I suggest professional English proofreading.
Response. We have added the suggested content to the manuscript in the respective parts of the manuscript. Additionally, we cited recommended article about modification of lncRNA in introduction and in the part about ASOs therapy. The English language was refined by a native English editor.
Round 2
Reviewer 5 Report
Comments and Suggestions for Authors
The extensive revision of the manuscript "m6A RNA editing: technologies behind future anti-cancer therapy" has contributed very much to its improvement. All my comments were satisfactorily addressed and implemented by the authors, so now I want to recommend this manuscript for publication.
Best regards
Author Response
We cordially thank the respected Reviewer for the valuable comments
Reviewer 6 Report
Comments and Suggestions for Authors
Dear Authors,
I see the changes you have made. In the "Introduction", a recent reference concerning m6A in lncRNAs and cancer should be cited: doi: 10.3389/fonc.2023.1063636.
Author Response
We thank the Reviewer for the contribution and improvement of our manuscript. The final correction was added as requested